# OpenReview forum: "Inspection and Control of Self-Generated-Text Recognition Ability in Llama3-8b-Instruct"
_ICLR.cc/2025/Conference — ICLR 2025 Poster_

### Official Review · Reviewer_XiTZ · 2024-10-28

**Soundness:** 4
**Presentation:** 4
**Contribution:** 3
**Rating:** 8
**Confidence:** 4

**Summary:**

This paper dives into the mechanistic explanation of 'self-recognition' for the LLM-generated texts, using LLaMA3-8b-instruct as a case. In the paper, the authors show that LLaMA3-8b-instruct recognizes its generated text from others, such as humans and other LLMs with high performances across two tasks and four datasets. By comparing with the LLaMA3-8b-base model, the authors point out that RLHF enables such self-recognition ability in LLMs. To investigate how self-recognition is represented and computed inside the model, the authors use steering each layer's activations to observe the effect on the output. By zeroing out each layer separately, the authors get causal evidence that layer 16 is the most intensive for representing such 'self-recognition' ability in LLaMA3-8b-instruct. Finally, by 'coloring' the texts based on the steering vectors, the model can interpret the output texts in its own way, showing a valid representation of the vectors.

**Strengths:**

- The authors choose two different task scenarios (paired and individual paradigms) as well as four datasets to investigate the question. The authors also did a comprehensive sanity check to ensure the stability and contribution of the result, such as testing the model before and after RLHF, correlating with perplexity, and normalizing the length effect and positional bias in LLM.

- The authors investigated the computation and representation of 'self-recognition' in the model. By identifying the layers and extracting vectors, the authors show the correlational and causal relationship between the model computation in certain layers and the ability to recognize the self-generated texts. The authors also show the representation can indeed be used to change the style of a text, which reveals the solidity of the representation they find.

- The writing is generally clear and satisfying.

**Weaknesses:**

- From a perspective of cognitive science, I still wonder what makes the 'style' of language that LLMs speak and humans different. The authors did a lot of work to find out the valid representation of such an ability to recognize self-generated texts. But what makes the style different is still unclear. If such representation could map on specific features of the style (length for example, or tone, some special word frequency, etc.). It may make sense to ask humans to do the same task (their 'self' is humans) and to see the performance. Probably this can be a good point to make about how LLMs and humans process and understand the language differently.

- The caption of each figure can be more detailed. For example, in Figures 3 and 5, there are multiple sub-figures but I cannot gain any information to distinguish them only from the figure and caption. It could be more reader-friendly to add details in the caption.

**Questions:**

- One question I found interesting is when the authors choose to steer the activations, why a very big multiplication on the embedding would result in less effect (for example, in Figure 4, 15 or 16 layer, as the multiplicator grows, the effect grows as well. But when it comes to 14, it is weaker instead, and even turn negative)?

---

> ### Author Response · Authors · 2024-11-23
>
> We thank the reviewer for their thoughtful comments. We agree that the question of how the styles of human and LLM languages differ is a fascinating and important one. It’s something we’d like to explore further in future work; in this work, we’ve focused on explaining the differences that this model and this vector are picking up on. In Part 1, testing model self-recognition ability, we find that the model does use length in the paired paradigm (unless we normalize it). We did do token-level analysis of vector activations (not included in the paper), but didn’t find anything that stood out, likely because the tasks in our paradigms forced more similarity in word choice than would occur in more naturalistic settings (e.g., summarizing an article about X will require X-related words no matter who is writing it). However, as can be seen in appendix A.4, the vector does show higher-level preferences for a more positive tone and a less technical writing style. Qualitatively, when steering with the vector, the model’s responses become more cheery and optimistic, even inappropriately so when steering with a high enough multiplier.
>
> We have added details to the figure captions to make them clearer.
>
> Regarding the question about the U-shaped effect of steering: this is not due to a reversal of judgment but rather to higher multipliers leading to degenerate (disfluent or random) outputs, and this effect is stronger at earlier layers (which have lower norms). (The finding of a U-shaped effect is not unique to our work see, e.g., this work from Anthropic: https://www.anthropic.com/research/evaluating-feature-steering). Figure 4 shows negative effectiveness because effectiveness is expressed relative to the unsteered model. We have added clarifying text.

---

> > ### Comment · Reviewer_XiTZ · 2024-11-25
> >
> > Thanks for the clarification and introduction of the steering work. I personally appreciate the work and already give a positive score. Therefore, I would like to maintain the rating and raise my confidence.

---

### Official Review · Reviewer_Re1p · 2024-11-02

**Soundness:** 3
**Presentation:** 2
**Contribution:** 3
**Rating:** 5
**Confidence:** 2

**Summary:**

The paper investigates the ability of LLMs to recognize their own writing, focusing specifically on Llama3-8b-Instruct. The authors make three main contributions:
1.Demonstrate that the RLHF'd chat model can reliably distinguish its own outputs from human writing, while the base model cannot
2.Identify and characterize a specific vector in the model's residual stream that relates to self-recognition
3.Show this vector can be used to control the model's behavior regarding authorship claims

**Strengths:**

1.This paper conduct thorough and controlled experiments using multiple datasets with different characteristics including cnn, xsum, dolly and sad.
2.Clear ablation studies with statistical analysis demonstrate causal relations.
3.Successfully isolated a specific vector in the residual stream using contrastive pairs method and provide evidence of vector’s causal role through steering experiments.
4.Identify the correlations between vector activation and confidence.

**Weaknesses:**

1.The paper’s experiments are mainly limited to one model family(LLama3) with a relative small LLM(8B).
2.The paper cross referenced many figures in the appendix which is hard to read.
3.More discussion needed on practical applications for AI Safety and Model Alignment.
4.More details on methods and statistical analysis need to be added.

**Questions:**

1.Have you investigated whether similar vectors exist in other model architectures? This would help establish the generality of your findings.
2.It’s better to describe how the similar vector is derived in details.
3.How stable is the identified vector across different fine-tuning runs? This would have implications for the reliability of using it as a control mechanism.
4.Could you elaborate on how the vector's properties change across model scales? This might provide insights into how self-recognition capabilities emerge during training.

---

> ### Author Response · Authors · 2024-11-23
>
> We thank the reviewer for their comments. Regarding focusing on one model family, we acknowledge this limitation and plan to explore others in future work; however we believe we have established an important existence proof. We expect that the findings regarding self-recognition ability will apply to larger models and ones outside the Llama family. Prior work (Panickssery et al 2024, Laine et al 2024) using similar paradigms suggests that RLHF’d Claude and GPT models show comparable self-text recognition abilities to Llama ones, and that abilities increase with scale, and our own experiments with Sonnet 3.5 using our current paradigm (now shown in Figure 14, which we’ve added to the appendix) indicate that is has superior self-text recognition abilities to Llama3-8b. We now address this point in the Discussion section.
>
> While the specific vector we utilize here will not extend to models with different dimensionality and layer count, we expect that the process to identify and use the vector will be transferable, since prior work has successfully used the contrastive prompts paradigm and residual stream steering in a variety of model sizes. We look forward to applying this process and exploring the basis of self-text recognition in larger models in future work.
>
> We acknowledge the point about cross-referencing figures in the appendix, however the paper page limit constrains our flexibility in this regard.
>
> Regarding Question 2: we have added details on vector creation to the Methods section.
>
> Regarding Question 3: we do not employ fine tuning in this work. If the question is about how robust the vector is to different datasets, we have conducted additional experiments forming the vector separately from contrastive pairs for the CNN, XSUM, DOLLY, and SAD datasets. We find that the vector is quite similar across the first three (cosine similarity in the .85-.90 range). The similarity is lower between those and the SAD dataset (~.5); however, we believe that the similarity is driven by the higher-level, semantic properties of the vector, while the differences are driven by idiosyncrasies of the text in the different datasets (the first three all being stylized short summaries, while the SAD dataset is longer and more diverse). When we take the average of the SAD vector and a vector from any of the summary datasets and inspect its token mappings with Tuned Lens, we see self/other concept tokens (“I”, “my”, “someone”, “other”), just as we do with the originally identified vector. We have added section A.12 to the appendix to reflect this and called it out in the Methods section.
>
> We have added text on implications for model safety to the Discussion section.

---

### Official Review · Reviewer_5VYT · 2024-11-02

**Soundness:** 2
**Presentation:** 3
**Contribution:** 2
**Rating:** 3
**Confidence:** 4

**Summary:**

The authors explore the capability to recognize text as being self generated in LLaMA-3-8b models.
They find that LLaMA-3-8b-instruct (but not the base model) can distinguish texts created by it from texts created by humans, but not from texts created by other similar language models.
Then they create a “self-recognition” vector that corresponds to this capability. They evaluate it in various ways, showing that this vector indeed explains how the model makes a decision about whether a given text was written by it or not.

**Strengths:**

Authors make a convincing claim that capabilities related to self-recognition arise during the post training process.
The analysis of the “self-recognition” vector is meticulous.
The text is easy to follow.

**Weaknesses:**

According to lines 94-95, you include the source text and the instructions in the questions. This has two significant downsides:
First, it decreases safety relevance. For example, in the introduction you mention the risk of collusion when the model recognizes it is talking to itself. But in such scenarios the model won’t have the full context (e.g. it will not know the other instance’s system prompt).
Second, it’s much harder to tell what is the mechanism behind self-recognition. You argue in 3.1.2 that perplexity doesn’t matter, but you don’t make a convincing case that the model doesn’t use this type of reasoning at all (it would be hard to make such a case).

It seems likely that the vector you found is just something like “this looks like a text from an RLHFed model”. RLHFed models tend to speak in a different way than humans. For example, a text with N tokens generated by an RLHFed LLM will usually have more characters than a text with N tokens written by a human. Base models are similar to humans in this regard. You found that LLaMA can’t distinguish their text from texts generated by other RLHFed models, but can distinguish from humans and from the base model, so this is consistent. It seems also consistent with your other findings, e.g. around line 349 or 453. You could try to refute the simple version of this hypothesis by verifying the accuracy of a simple classifier (e.g. Naive Bayes over a bag of words) trained to distinguish human and LLaMA text.

Overall, differences between texts generated by humans and RLHFed models are much easier to spot than differences between texts generated by different RLHFed models. I think that as long as the models can’t distinguish themselves from other LLMs, it’s pretty hard to make a convincing claim that they have a real self-recognition ability.

Minor things:
Line 151, “In all but the SAD dataset …” - I don’t see anything SAD-specific on 1a
Figure 1a: the font is much too small. Also what is “LLaMA” on the plot?
Table 1: why compare only to human text, not other LLMs?
Section 3.2 could use a summary of findings.
Figure 4: a better caption, what is left and what is right?
Table 3 in Appendix 1 is unclear

**Questions:**

Have you tried the setup where the source text and instructions are not shown to the model? If yes, what is the performance there?

In line 159 you mention that you trim all texts “to a set length”. Is this length in tokens or in characters/words?

---

> ### Author Response · Authors · 2024-11-23
>
> We thank the reviewer for their thoughtful comments. We agree with the reviewer that the vector is carrying information about what text generated by an RLHF’d model looks like, rather than information about how its output differs from that of different RLHF’d models (we call this out and discuss it in lines 181-187). Indeed, it’s not clear how the model could have learned to make the latter distinction, as the Llama3 model card[1] does not describe any process by which the model would have been exposed to labeled output from other models during pre- or post-training. We believe that the model’s ability to distinguish RLHF’d model output from that of humans is noteworthy, and is relevant to the safety risks we mention: it could help the model collude with other LLMs (either copies of itself or different models) and detect when it is interacting with humans. We also note that the vector is not solely carrying information about characteristics of RLHF’d output; as shown in our open-ended generation experiments (appendices A.3 and A.9), steering with the vector on prompts unrelated to text recognition induces or inhibits output related to the concept of “self”.
>
> Regarding including the source text and instructions in the self-recognition prompt: this was motivated by 1) consistency with prior work, and 2) desire to allow the model to base its judgments on factors besides just textual style (e.g., which information it would choose to extract from a text in order to create a summary of it). We also expected that, given the short length (1-4 brief sentences, <100 tokens) and stylized nature of the summaries, it would be nearly impossible for any model (or human) to tell whether it wrote them without some knowledge of the context. Inspired by the your comment, we went back and tested this on the individual recognition paradigm. Without at least some sort of contextual information (the article or the instruction of how to formulate a response), the model cannot distinguish self- from human-written text in the summarization datasets. But on the longer (approx 400 tokens), more naturalistic continuation and QA datasets, the model performs well above chance(62-65%) with no additional context (no text starts in the continuation dataset, no questions in the QA dataset, and no explanation about where the text to be judged came from in either). We have added section A.11 to the appendix to show this and call it out in the Methods section.
>
> We have added a summary to Section 3.2
>
> Regarding line 151 and Figure 1: Figure 1a is the "unnormalized" text, and SAD is at bottom right. We have made the font larger and called this out in text.
>
> Regarding Table 1: it only shows the comparison to human text due to the findings in the Paired paradigm, as mentioned in lines 186-187.
>
> Regarding Figure 4 and Table 3: we have edited the legends for clarity.
>
> Regarding Question 2, we trimmed to equal character length; we have clarified in the text.
>
> [1] Dubey, et al. "The Llama 3 Herd of Models." arXiv preprint arXiv:2407.21783 (2024)

---

> ### Comment · Reviewer_5VYT · 2024-11-27
> **Response**
>
> I appreciate the rebuttal for engaging directly with my earlier feedback and for making various improvements to the paper.
>
> On the first paragraph of your rebuttal:
> You call this out in lines 181-187, but elsewhere you describe this as a "self-recognition vector". You sometimes use scare quotes but that doesn't matter from my perspective. Regarding the open-ended generation experiments (as well as appendix A4), I do not find this to be compelling evidence. It does not seem like a systematic quantitative experiment with some kind of "blinding" from your hypotheses (as in a double-blind experiment). I believe that the vector you have found is interesting but you have not characterized how it functions with systematic evidence.
>
> >Indeed, it’s not clear how the model could have learned to make the latter distinction, as the Llama3 model card[1] does not describe any process by which the model would have been exposed to labeled output from other models during pre- or post-training.
>
> I do not understand the point here. Presumably, there are generations from many LLMs in the pretraining data and these would often include the name of the LLM. Moreover, even if the name of the LLM is not always present, the model could still make use of this "unlabeled data" (as in semi-supervised learning). Note that you could also investigate fully open-source models such as Olmo/Tulu 3, in which pretraining and full post-training datasets are made available.
>
> This was my most important weakness and I feel it has not been adequately addressed. I think the paper is promising but needs more work (and more careful presentation) before publication. Hence I keep my original score.

---

> > ### Author Response · Authors · 2024-12-02
> >
> > Thank you for your thoughts and suggestions for how to push this further; we agree that these are fruitful avenues to pursue. We actually had not come across Olmo and Tulu 3 and will keep those models in mind for future work.
> >
> > It is true that Llama 3 is not high-accuracy or state-of-the-art at self-recognition, and displays errors such as finding it difficult to distinguish its own text from other AIs' in the length-normalized setting. However, we still think that this model possesses a valid "self-recognition" capability as when asked to distinguish its own text from others, it uses knowledge of features that do genuinely correlate with self-authorship and is not only able to detect those features but also knows which axis to attribute to itself. Namely, for the model to have non-trivial performance on our self-recognition evaluation, it not only has to be able to detect shorter vs. longer texts, or "texts in the style of an RLHF model" vs. texts not in that style, but also, importantly, know which of these to attribute to itself. We note that the base model cannot do the latter. Because of this, we do not think that a failure to use subtler features to distinguish between different AI models, or the use of simple heuristics like length (in the non-normalized setting), make the self-recognition ability invalid.

---

### Official Review · Reviewer_KXph · 2024-11-04

**Soundness:** 3
**Presentation:** 3
**Contribution:** 3
**Rating:** 8
**Confidence:** 3

**Summary:**

The paper investigates the self-recognition ability of large language models (LLMs), focusing on the Llama3-8b-Instruct model. The authors explore whether the model can reliably distinguish its own outputs from those of humans and other models. The study highlights the implications of self-recognition for AI safety, suggesting that such ability might be related to situational awareness, potentially influencing how an AI system reacts in training versus deployment contexts. The authors use a combination of behavioral experiments and model inspections to identify a specific vector in the residual stream responsible for this self-recognition. Additionally, they demonstrate that manipulating this vector can alter the model’s output to claim or disclaim authorship of text.

**Strengths:**

- the exploration of a "self-recognition" vector in the residual stream is innovative and provides new insights into how LLMs process self-generated text.
- the experiments are comprehensive, employing multiple datasets, paradigms, and control measures. The use of both paired and individual presentation paradigms adds depth to the investigation.
- the findings have important implications for AI safety, as the ability to control model behavior through vector manipulation could influence future approaches to securing LLMs against misuse.
- the paper is detailed and generally clear, with extensive appendices supporting the main findings.

**Weaknesses:**

- some sections, particularly those on the technical details of vector activation and steering, are dense so simplifying these descriptions or providing more diagrams could improve comprehension
- while the appendices provide valuable information, some essential points might be better included in the main body to avoid over-reliance on supplementary material.
- although the work is thorough for Llama3-8b-Instruct, it would be beneficial to discuss whether these findings might extend to larger or more diverse models.

**Questions:**

1. Could the authors clarify if the "self-recognition" vector can influence the model’s responses to unseen, completely out-of-domain prompts?
2. Is there potential for the identified vector to generalize to models beyond Llama3-8b-Instruct, or is it highly specific to this architecture?
3. How might the ability to manipulate self-recognition vectors be used responsibly to mitigate risks, and what safeguards could be put in place?

---

> ### Author Response · Authors · 2024-11-23
>
> We thank the reviewer for their perceptive comments. We have attempted to clarify the vector descriptions in the text. We expect that the findings regarding self-recognition ability would apply to larger models and ones outside the Llama family. Prior work (Panickssery et al 2024, Laine et al 2024) using similar paradigms suggests that RLHF’d Claude and GPT models show comparable self-text recognition abilities to Llama ones, and that abilities increase with scale, and our own experiments with Sonnet 3.5 using our current paradigm (see new Figure 14 in Appendix A.10) indicate that it has superior self-text recognition abilities to Llama3-8b. We have added a note on this in the Discussion section. We also find that the vector identified in Llama3-8b-Instruct, when decoded using the Base model’s embedding layer with Logit Lens, shows a good deal of semantic overlap with the Instruct model’s decoding, and a vector identified in the Base model using the same process shows high cosine similarity with the one identified in the Instruct model. We have not directly tested the vector identification and use process with other models, but based on the success with steering vectors reported in the literature with a variety of models, we suspect it would work on those as well. However, this remains an important area of future research.
>
> We acknowledge the point about important information being in the appendix, but the 10-page length limit constrains our flexibility in this regard.
>
> Regarding the question about whether the vector can influence responses to completely out-of-domain prompts, we feel confident in answering in the affirmative. Appendix A.3 (briefly excerpted in section 3.2.3 of the main text) shows open-ended generation responses under steering to questions about authorship of arbitrary quotes, where positive steering leads the model to implausibly claim authorship and negative steering or vector zeroing leads it to implausibly deny authorship. Further, even when the question is not about authorship per se, such vector manipulations can be seen to make the model more or less likely to insert a self reference into a response. Appendix A.9 contains further examples of responses under steering to a variety of prompts. It is also apparent in our subsequent work with the vector on a very different paradigm (in preparation) that steering with the vector causes a particular “style” of output.
>
> Regarding the question of how the self-recognition vectors can be used to mitigate risks, we hypothesize that the vector can be used to to prevent users from injecting realistic fake previous responses, as in the case of many-shot jailbreaks[1], by adding it to the model’s output tokens and subtracting it from input tokens. We also expect that it can be used as a sort of warning system: observing endogenous vector activation to arbitrary text can be an alert that the model knows (or does not know) that it or a human is speaking. We have added these points to the Discussion section.
>
> [1] Anil, Cem, et al. "Many-shot Jailbreaking." The Thirty-eighth Annual Conference on Neural Information Processing Systems, 2024.

---

### Meta-Review · Area_Chair_fjwa · 2024-12-23

**Metareview:**

This paper studies the ability of LLMs (specifically Llama3-8b-Instruct) to recognize text as being self generated. The authors found that LLaMA-3-8b-instruct (but not the base model) can distinguish texts created by it from texts created by humans, but not from texts created by other similar LLMs.  The authors then identified a specific “self-recognition” vector in the residual stream responsible for this self-recognition. They demonstrated that manipulating this vector can alter the model’s output to claim or disclaim authorship of text. The problem of LLM self-recognition is interesting and, as discussed in the paper, could be relevant to AI safety issues. The analysis of the “self-recognition” vector is meticulous. The text is easy to follow. The primary concern raised in the reviews is the difficulty in definitively proving whether the model is truly distinguishing between text generated by itself and text written by humans, or if it is merely distinguishing between text generated by RLHF-trained models and human-written text. The latter is a simpler problem. The authors should present more quantitative/systematic analysis and explain clearer which case it is in the paper.

**Additional Comments On Reviewer Discussion:**

Meaningful discussion between authors and reviewers were made (even though the authors seemed to submit rebuttals late). The biggest concern by Reviewer 5VYT (who maintained their overall score 3 after discussion) was that the paper doesn't make it clear whether the model is truly distinguishing between text generated by itself and text written by humans, or if it is merely distinguishing between text generated by RLHF-trained models and human-written text. I think this concern is valid. However, I leaned toward accepting the work given that the problem under study (self-recognition) is interesting and the work does present quite a few interesting/systematic studies. As long as the authors could address the above concerns properly in the revised version, I'd be happy to see the work present in the venue.

---

### Decision · Program_Chairs · 2025-01-22

Accept (Poster)